# Attitude of Midwives towards Fluoride Recommendations and Oral Prevention in Infants and Young Children

**DOI:** 10.3390/children9081135

**Published:** 2022-07-29

**Authors:** Antje Geiken, Louise Holtmann, Christof E. Doerfer, Christiane Schwarz, Christian Graetz

**Affiliations:** 1Clinic for Conservative Dentistry and Periodontology, University Schleswig-Holstein, 24105 Kiel, Germany; holtmann@konspar.uni-kiel.de (L.H.); doerfer@konspar.uni-kiel.de (C.E.D.); graetz@konspar.uni-kiel.de (C.G.); 2Institute of Health Sciences, Department of Midwifery Science, University of Luebeck, 23562 Luebeck, Germany; christiane.schwarz@uni-luebeck.de

**Keywords:** early childhood caries, primary teeth, fluoride, caries prevention, midwives

## Abstract

Early childhood caries is a challenge. Early dental screening flanked by multidisciplinary preventions by pediatricians, dentists, and midwives (MWs) may be helpful. New recommendations for dental screening in children (FUs) and fluoride have been introduced in Germany. The aim of this study was to investigate whether midwives consider FUs useful and implement early childhood caries prevention, as well as fluoride recommendations. The survey was conducted using an online questionnaire. Demographic data, including 11 items on early childhood dental prophylaxis and fluoride, were requested. Agreement was recorded using Likert scales. The data were analyzed descriptively. Two hundred and seventeen female MWs participated (age: 44.1 (11.04) years). One hundred and four (47.9%) participants knew about the FUs. Of the MWs, 30.7% found a referral from the first tooth to be very important (important/neutral/unimportant: 27%/27.9%/14.4%), compared with 84.8% for the entire primary dentition (11.8%/2.8%/0.5%). Of the MWs, 41.7% always recommended fluoride toothpaste from the first tooth (often/occasionally/rarely/never: 22.7%/12.4%/7.9%/15.3%) and 48.1% completely rejected fluoride-free toothpaste (always/often/occasionally/rarely: 9.8%/8.9%/17.3%/15.9%). In addition, 54.8% never recommended the use of fluoride tablets (always/often/occasionally/rarely: 9.2%/7.4%/10.2%/18.4%). The FUs are not yet well-known among MWs, and only less than one-third recommended dental check-ups, starting with the first tooth. This contrasts with the high uptake of fluoridated toothpaste. More educational work should be carried out to convince more MWs of the benefits of the FUs.

## 1. Introduction

Pregnant women show a high acceptance rate for seeking prenatal care [1]. The main purpose of prenatal care is the consultation and monitoring of pregnancy by a doctor or midwife. Its aim is to detect deviations from the normal course of pregnancy as early as possible. As mothers are open-minded toward health education, especially when received from midwives, their involvement and expertise in dental preventive strategies can have a promising impact. It is known that early prevention using motivational interviewing and health counseling, as well as integrating ECC prevention into mainstream healthcare, can be a promising approach [2]. This makes prenatal care an excellent, low-threshold approach for communicating information about early childhood dental health. Because early childhood caries (ECC) is still an unsolved problem, it is widely considered to be the most common chronic disease in children [3]. All of the medical professional groups involved, including pediatricians and dentists, but also midwives, can make valuable contributions to the prevention of caries. The treatment is time-consuming and challenging; due to the limited cooperation of the children, outpatient rehabilitation may not be possible, and a more cost-intensive and risky treatment under general anesthesia is, therefore, often necessary [4]. However, ECC does not only affect the teeth. During its progression, it also causes inflammation, abscesses, and as secondary consequences, sleep disturbances, concentration problems, and speech disorders [5,6,7]. Therefore, ECC must be considered as a disease that limits the quality of life in general [7]. Furthermore, affected children have an increased risk of developing caries in their permanent dentition as well [8]. The known risk factors are a low social status, inadequate oral hygiene, and a poor cariogenic diet. Early detection is one of the essential parts of prevention. Dental examinations in early childhood could detect caries in the primary dentition at an early stage and counteract social polarization. The late presentation of children with a high risk of caries to a dental practice often means that this group of patients has already developed carious lesions that need to be treated urgently.

Mothers are considered the main carriers of the oral microbiome, but also of cariogenic germs [9]. The earlier this colonization of the child’s oral cavity takes place and the higher the microbial load of the maternal oral cavity observed, the higher the risk for the development of ECC [10]. The children of mothers with an increased level of caries-relevant bacteria have more carious defects than the children of mothers with a lower bacterial load [11]. Unfortunately, not enough pregnant women attend dental check-ups or are aware of the importance of oral health [12,13,14]. In addition, parents of young children do not have sufficient knowledge about nutrition in relation to the consumption of sugar and sticky foods, as well as oral hygiene [15,16,17]. Early prevention and the education of parents could, therefore, also counteract the development of ECC during pregnancy [18]. Studies show promising results in this regard [13,19,20,21].

In Germany, new dental screening examinations for children were introduced in 2019 in order to lower the general risk of ECC in the German population. The earliest possible examination by a dentist from the deciduous tooth onwards is considered very sensible in order to avoid the development of ECC, and should be aimed for in this young patient group [22]. This screening has been possible from the first deciduous tooth since 2019 (FU1a, sixth months of life) [23]. Further dental check-ups can be carried out at the ages of 10–20 months (FU1b) and 21–33 months (FU1c). It is also possible to provide oral hygiene instructions to parents at the dentist’s office and have them practice brushing their teeth on the spot. In addition to oral hygiene training and dental diagnostics, a fluoride varnish can be applied twice a year (every six months).

Another undisputed factor in caries prevention is fluoride. In general, fluoridation is possible via local or systemic application [24]. In the past, the fluoride guidelines of dentists and pediatricians were different and contradicted each other in Germany [25,26].

The associations of pediatricians (the German Society for Pediatrics and Adolescent Medicine (DGKJ), in addition to the German Academy for Pediatrics and Adolescent Medicine (DAKJ)), rejected the use of fluoridated toothpaste if the children had not yet sufficiently mastered spitting out toothpaste (mostly learned by the age of five–six). Instead, the administration of fluoride tablets from birth was recommended [25]. In contrast, dentists recommended brushing teeth with fluoridated toothpaste from the first deciduous tooth and not providing fluoride tablets [26].

In the past, these inconsistent guidelines led to confusion and misunderstandings among parents, midwives, dentists, and pediatricians about the correct use of fluoride. In 2021, commissioned interdisciplinary fluoride recommendations from dentists and pediatricians were finally published, facilitating the standardization of the use of fluoride [27].

Midwives are often the first confidants of and medical contacts that care for expectant mothers during pregnancy, birth, and the first months of the child’s life. They conduct close consultations and examinations of pregnant women. In this way, they provide important impulses, can communicate prevention approaches [28], and also have a major responsibility in educating parents about oral health, i.e., healthy eating, tooth brushing, fluoride use, and dental recommendations.

This means that nondental staff can also be involved in prevention work, which saves resources and costs. Azevedo et al. [29] showed that nondental personnel can also achieve promising results in preventing the prevalence of caries by 12.9%.

As far as the authors are aware, no study has yet been conducted on the extent to which midwives are informed about the new dental screening examinations and fluoride recommendations in Germany.

Therefore, the aim of this study was to investigate whether or not midwives are aware of the new dental check-ups and consider them useful, to what extent they support the fluoride recommendations, and which fluoride applications they recommend to parents.

## 2. Materials and Methods

### 2.1. Study Population

This study followed the current guidelines of the 2013 Declaration of Helsinki (Fortaleza, Brazil), and the Ethics Committee of Kiel University approved this study (AZ: D452/18).

In 2021, all 16 regional midwifery associations in Germany, which are members of the umbrella organization of the German Association of Midwives (DHV), were contacted and asked for their support in conducting the study survey. Eight midwifery associations (Bavaria, Baden-Wuerttemberg, Schleswig-Holstein, Bremen, Brandenburg, Thuringia, Hesse, and North Rhine-Westphalia) agreed to advertise the survey to their members. A total of 13,220 members were registered in these 8 associations and could have participated in the online survey via study calls. The study was conducted digitally, which means that no paper-based questionnaires had to be distributed by mail or at a conference or meeting. Instead, an online questionnaire served as the survey instrument. A cover letter, an introductory text, a survey link, and a QR code were sent to the participating regional midwifery associations. The questionnaire was accessible from May 2021 to September 2021. Participation was voluntary and no financial or other incentives were promised. Everyone was informed about the content of the study/questionnaire and consented to participate. Due to the anonymous questionnaire design, nonrespondents could not be contacted to increase motivation to participate. Retired midwives and midwives, who were not practicing but were in university/clinical settings or public service, were excluded from participation, as were students and trainees. No questionnaires were taken into account that were terminated prematurely, i.e., after the introductory page, if there was no indication of gender and place of occupation (federal state), or questionnaires with incorrect information on age, gender, occupation, or federal state, such as age xx or 00 years.

### 2.2. Questionnaire

The questionnaire was validated by six experts in midwifery science, dentistry, and web-based surveys at the University of Kiel and the University of Luebeck, and was converted into an online survey using a web-based survey tool (Unipark, QuestBack GmbH, Cologne, Germany).

A pretest of the questionnaire was conducted in a focus group of 30 midwives.

Eleven items in total were asked. The questionnaire consisted of 14 questions and was divided into 2 sections. In the first section, the demographic data of the participants were collected, e.g., activity in a private practice, age, gender, and place of residence (federal state), and the second section asked about awareness of the FUs, the implementation of caries prevention measures in children under 33 months of age (ten items), and about fluoride recommendations (one item, divided into seven subitems) (Table 1). Likert scales were used to rate the answers, e.g., from 1 to 4, “very important”, “important”, “neutral”, and “unimportant”, or 1 to 5, where 1 stands for “never” and 5 for “always”. There was no obligation to answer all of the questions. Filling out the questionnaire took about eight minutes.

### 2.3. Statistical Analysis

For the statistical analysis, the data were entered into SPSS Statistics for Mac (SPSS Statistics 24, IBM, Chicago, IL, USA). The analysis of the data was primarily descriptive, with the percentage frequencies, means, and Likert scales reported as medians with reference to the lower/upper quartile. The significance level was defined at 95% of statistical probability (*p* < 0.05). Graphs were created using Microsoft Excel 2011 for Mac (version 14.3.2, Microsoft Corporation, Redmond, WA, USA).

## 3. Results

There are a total of 13,220 members registered with local midwifery associations who could have participated in the study. The questionnaire was accessed 1702 times, with 1342 accesses being cancelled after the introduction page. Finally, 217 midwives participated in the survey (Figure 1). The response rate was 1.64%.

The participants were, as a mean (SD), 44.1 (11.04) years old and exclusively female (Table 2).

The FUs were known to 104 (47.9%) midwives, and 113 (52.1%) were not aware of them. In terms of the FUs, 50 (23%) midwives informed all parents about them and recommended visiting a dentist, 62 (28.2%) only provided information about the possibility of FUs without providing further information or recommending a visit to a dentist, 51 (23.5%) only informed parents about them if they seemed interested, and 54 (24.9%) did not inform parents about FU dental examinations at all. Three participants did not answer the question.

The majority of the participants (163/75.1%) found education about FUs not very burdensome (very burdensome 11/5.1%, burdensome 43/19.8%).

Dental presentation from the first tooth was considered very important by 66 (30.7%) of the midwives surveyed and important by 58 (27%). With the increasing age of infants, midwives considered presentation more appropriate for FUs, 98 (46.5%) respondents considered presentation very important between 6 and 24 months, and 179 (84.9%) considered presentation very important for complete primary dentition (Table 3).

The question about the appropriate time to educate parents about FUs clearly turned out to be a postpartum issue, for example, after birth or during breastfeeding. Thus, 157 (74.1%) of the midwives responded that education was very important during breastfeeding, 88 (41.7%) after birth, and only 30 (15.1%)/32 (16.1%) during prenatal care/pregnancy (Table 4).

### Specific Caries Prevention Scenarios and Fluoride Recommendations for Children between 6 and 33 Months of Age

All 217 of the participants answered the set of questions on “education on caries prevention”, “education on oral hygiene measures”, and “nutritional counselling in connection with caries prevention” as valid.

The majority of the respondents always (*n* = 106/48.8%) and often (*n* = 55/25.3%) educated parents about caries prevention. In contrast, only 16 (7.3%) of the participating midwives always informed parents about oral hygiene measures for children. A small number of the midwives, 16 (7.3%), always conducted training on oral hygiene measures with the children’s parents, while the majority (*n* = 70/32.3%) did not do so at all. A majority of 90 (41.5%) participants reported always advising the children’s parents about nutrition, with only 11 (5.1%) doing so rarely and 13 (6%) not doing so at all (Table 5).

For the question “Do you recommend fluoride toothpaste from the first tooth?”, 216 valid responses could be evaluated, 213 to “Do you recommend fluoride toothpaste when the child can spit?”, 214 to “Do you recommend fluoride-free toothpaste?”, and 217 “Do you recommend fluoride tablets?”. Fewer than half (*n* = 90/41.7%) of the participating midwives adhered to the current dental fluoride recommendations and always recommended fluoride toothpaste for children, but only 33 (15.3%) did not recommend it at all and completely rejected its use. Thirty-two (15%) of all of the participants recommended the use of a children’s toothpaste only if the child can spit it out, while the majority (*n* = 101/47.4%) did not. In contrast, the majority (*n* = 103/48.1%) declined to always recommend fluoride-free toothpaste, while only 21 (9.8%) midwives responded as recommending it (Table 6). Fluoride tablets were recommended.

## 4. Discussion

The recommendation for the first visit to the dentist was adapted to the international dental recommendations [30]. In Germany, parents now have the option of visiting a dentist with their child as early as when the first deciduous tooth reaches the oral cavity. This has a positive effect on children’s adherence and prevents fears from the very beginning. Midwives are important confidants for women during pregnancy and after birth. It has been scientifically proven that midwives have a strong impact on implementing appropriate healthcare behavior [24]. Therefore, they have an important, key position in the context of the new dental check-ups. Thus, midwives are to be regarded as important partners for oral health, but are not sufficiently known for this.

### 4.1. Caries Prevention and Oral Hygiene

Unfortunately, half (52.1%) of the midwives surveyed were not aware of the new FUs. One has to wonder if the dissemination of information by the dental association cannot be considered sufficient and if midwives are perhaps not yet recognized as important partners in children’s oral health. This aspect could be taken up in the context of further research questions.

However, the information provided by the FUs is also not passed on to all parents by the participants; 24.9% said that this information was not passed on.

A recent study from France asked 494 health professionals, including 217 midwives, about their perspective on ECC prevention [31]. In this French study, only 34% of midwives taught parents about oral prevention. One reason for this difference could be that the majority of midwives in this study were not aware of the symptoms of ECC and were not sufficiently sensitized to the clinical picture of ECC. However, there seems to be an interest in oral prevention among midwives. This also seems to be confirmed by Tourino, et al. [32]. In the mentioned study by Tourino, the vast majority (82.0%) of student midwives affirmed that oral health education was a useful part of the curriculum. Midwives who have already completed their training also asked for further education and training on oral health [33,34]. A lack of understanding of ECC seems to be contradicted by other studies. ECC was known to 86% of the midwives surveyed in the study by Ehlers et al. [35].

The results of our study show that the majority of the respondents consider going to the dentist from the first tooth as very important or important (57.7%). These are slightly higher approval ratings than in previous studies. For example, Rahman et al. [36] showed that 45.6% would recommend dental presentation from the first tooth, compared to only 28.6% in the study by Ehlers et al. [35] and only 5.7% in the study by Wagner and Heinrich-Weltzien [33]. Therefore, the results of this study can be considered a positive development in the course of time. In all of the previous studies, and also in the present questionnaire, it is common that consent to dental presentation increases with the age of the child [33,35,37]. However, it is not only midwives who seem to think that going to the dentist is not necessary from the very first tooth. Other nondental professionals, such as pediatricians, also rarely recommend it [37,38,39]. The majority of European pediatricians (43%) recommended a first visit to the dentist from the age of three years, and only 7% recommended a visit for children under one year of age [37].

In contrast, the midwives in this study were well-aware of their responsibility for oral prevention. The majority of the respondents always educated parents about nutritional counseling (41.5%) and caries prevention (48.8%), and 48.8% educated parents about oral hygiene measures. Other studies also confirmed that midwives feel obliged to educate about oral health [35].

However, in addition to education, practical guidance should also be given, as theoretical education is not sufficient to actually persuade parents to change their behavior [40]. Unfortunately, this practical implementation of oral hygiene training was insufficiently carried out by the participating midwives; only a minority of 7.4% answered that this was always implemented. Especially in this context, cooperation with dentists should be aimed for.

Usually, dentists have the possibilities and facilities to carry out practical oral hygiene education with parents in a child-friendly way.

### 4.2. Fluoride Recommendations

Fluoride is considered a crucial aspect in reducing tooth decay [41,42]. The caries prophylactic effect is not disputed by any medical group, but in the past, there have been disagreements in Germany about the methods of administration. On the one hand, pediatricians have mainly recommended systemic administration via tablets and brushing teeth with fluoridated toothpaste after 6 years of age, when children are supposed to safely spit it out [25,43]. Dentists, on the other hand, have recommended brushing with fluoridated toothpaste from the first deciduous tooth [26]. As already mentioned, these guidelines were in conflict and misleading, not only for pediatricians, dentists, and midwives, but mainly for parents. According to the majority of European and other international recommendations, systemic fluoride supplementation is of secondary importance compared to local fluoridation. It is, therefore, important that the currently common consensus recommendations on fluoride use in accordance with the international standards are published by pediatricians and dentists [27].

In our survey, the majority of the participating midwives always or often recommended the use of a fluoride toothpaste from the first deciduous tooth (always: 41.7%/often: 22.7%, collectively 64.4%), which is a high level of agreement when taking into account the discordance between pediatricians and dentists in the past. Previous studies showed lower values (55.3%).

A special aspect of this study compared to the previous studies is the change in fluoride recommendations.

On 21 April 2021, new consensus fluoride recommendations were published by dentists and pediatricians in Germany. In contrast to the previous dental recommendations, the administration of fluoride tablets is now recommended up to the first tooth. After that, fluoride tablets or a fluoridated toothpaste are recommended until the first year of life. After the first year, only the use of a fluoride toothpaste is recommended.

It appears that the dissemination of the new consensus recommendations on fluoridation, at least as far as the use of fluoridated toothpaste is concerned, has been more successful than dental examinations, and that the topic of fluoridation is given a high priority by midwives. It remains to be seen to what extent the new recommendations will be implemented and whether midwives will implement the recommendations to use fluoridated tablets in the first year. These questions should be re-examined in future studies.

### 4.3. Limitations of the Survey

The response rate of 1.61% seems rather low. However, it has to be taken into account that this study was planned as a full cohort investigation of all German midwives.

Sampling was performed through the midwife professional associations, of which 8 out of 16 agreed to support our project, leading to a first level of risk of bias. These associations posted the link/QR code to the questionnaire on their homepages, which again selected those who visit these homepages from those who do not. This is a second level of bias, but also explains the low response rate compared to evaluations at scientific meetings. In accordance with the General Data Protection Regulation (DGSVO 2018), participant demographic data were only collected to a limited extent to avoid subsequent identification.

Nevertheless, this purely internet-based questionnaire had certain advantages. One is that transmission errors from paper form to the data base are excluded, which is an important quality instrument. The second is that the online participants did not feel observed and completed the questionnaire with less social pressure and under anonymous conditions (data protection). Due to the recruitment of participants via the Internet, the current survey, as is the case with all online surveys, is subject to bias [44]. Thus, it is possible that midwives without Internet access could not participate in the studies. However, the proportion of the German population without Internet access can be considered very low [45]. The self-selection aspect is certainly critical. Subjects who voluntarily participate in online surveys may be more interested in the subject, which may have an influence on the results [46,47]. The participating midwives may have had a particular interest in the topics of caries prophylaxis, fluoridation, and accountability, and therefore responded with a higher motivation than the national average of midwives. Additionally, the questionnaires might have been answered unconsciously or consciously according to the wishes of the questionnaire creator, such that social desirability cannot be excluded. A sensitivity analysis of the nonresponders was not possible, due to the chosen anonymized study (approved by the local ethics committee), design and German data protection guidelines. It was not possible to contact midwives who did not respond to the call or participants who dropped out after the introductory page.

Therefore, we do not claim representativity and we did not carry out a quantitative analysis. Comparable studies reported different response rates. Nancy et al. [31] did not report a response rate, but reported the absolute number of 217 participants. Rahman et al. [36] had response rates of 26.6% and 42.3%; in absolute numbers, these were 149 and 338 participants, respectively. However, the study design differed from the method chosen by the authors. The survey took place during a congress, which allowed for a direct approach and a higher willingness to complete the questionnaire if necessary. Ehlers et al. [35] also reported a higher response rate (46%). This study also used a different methodology than the authors. A total of 503 people were contacted by mail and asked to participate; 283 took part in the survey, a number in absolute terms similar to the authors’ survey. Future studies should, therefore, be conducted with larger case numbers to reduce potential bias.

Based on comparisons with previous studies, the current study provides insights into the evolution of midwives’ attitudes and highlights the importance of involving midwives in oral health strategies in early childhood.

## 5. Conclusions

Midwives should be viewed as important disseminators in the field of oral prevention and fluoridation. They themselves seem to be well-aware of their role. Thus, the majority of the respondents always provide information about caries-preventive measures in nutrition and oral hygiene. However, the active training of parents by midwives in oral hygiene rarely takes place. The recommendation of taking a child to a dentist from the first deciduous tooth needs improvement. This should also be considered against the background that not all of the midwives interviewed were aware of the new FUs.

Therefore, it can be concluded that midwives should be more trained in and sensitized more to the FUs.

## Figures and Tables

**Figure 1 children-09-01135-f001:**
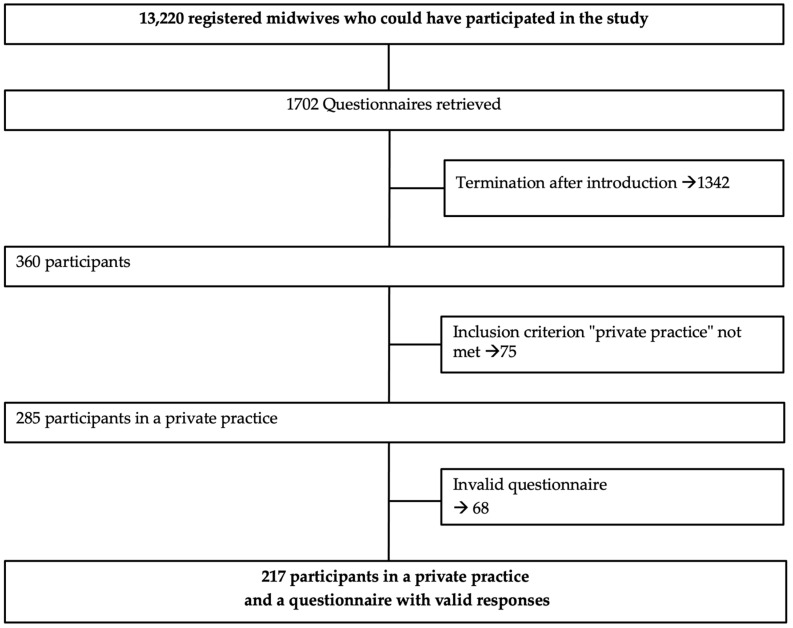
Recruitment scheme of the participating midwives in private practices.

**Table 1 children-09-01135-t001:** Second section of the questionnaire with survey questions and answer.

Survey Questions	Response Options
Are you aware of the new dental check-ups (FU1a (6–9 months), FU1b (10–20 months), FU1c (21–33 months))?	Yes No
Do you inform parents about the dental check-ups (FU) for children under 33 months? Please select.	All parents with children under 33 months are informed. All parents with children under 33 months are informed and recommended to the dental practice. Only parents who ask or show interest will be informed. No information is given in this context.
How time-consuming do you think it is to educate parents about FU? Please select.	Not very burdensome Very burdensome Burdensome
How useful do you think a presentation is in a dental office? From the first tooth Parallel to U5-U7 (6–24 months) From the complete first dentition Please mark only one cross per line.	Very important Important Neutral Unimportant
When do you think would be a good time to educate the parents about the FU? During pregnancy During birth preparation During the postpartum period During the breastfeeding period Please only one cross per line.	Very important Important Neutral Unimportant
To what extent do you do the following activities with children younger than 33 months? Caries prevention education Education on oral hygiene measures Oral hygiene training Nutritional counselling in connection with caries prevention Please only one cross per line.	Never Rare Occasionally Often Always
To what extent do you do the following activities with children younger than 33 months? Do you recommend fluoride toothpaste from the first tooth? Do you recommend fluoride toothpaste when the child can spit? Do you recommend fluoride-free toothpaste? Do you recommend fluoride tablets? Please only one cross per line.	Never Rare Occasionally Often Always

**Table 2 children-09-01135-t002:** Demographic data of the participants.

Demographic Data	Result
Number of participants	217
Age (years) (SD) (range)	44.1 (11.04) (24–68)

**Table 3 children-09-01135-t003:** Perceived significance of early dental visit by age group in absolute values and in percentages (%).

	Total	Very Important	Important	Neutral	Unimportant
From the first tooth	215	66	58	60	31
		(30.7%)	(27%)	(27.9%)	(14.4%)
Parallel to the U5–U7 (6–24 months)	211	98	71	34	8
		(46.5%)	(33.6%)	(16.1%)	(3.8%)
From the complete first dentition	211	179	25	6	1
		(84.9%)	(11.8%)	(2.8%)	(0.5%)

**Table 4 children-09-01135-t004:** Results regarding when the right time would be to educate parents about the FUs in absolute values and in percentages (%).

	Total	Very Important	Important	Neutral	Unimportant
During pregnancy	199	32	40	74	53
		(16.1%)	(20.1%)	(37.2%)	(26.6%)
During birth preparation	199	30	61	62	46
		(15.1%)	(30.7%)	(31.1%)	(23.1%)
During the postpartum period	211	88	74	32	17
		(41.7%)	(35.1%)	(15.1%)	(8.1%)
During the breastfeeding period	212	157	45	10	0
		(74.1%)	(21.2%)	(4.7%)	(0%)

**Table 5 children-09-01135-t005:** Measures carried out in children between 6 and 33 months as part of caries prevention in absolute values and in percentages (%).

	Total	Never	Rare	Occasionally	Often	Always
Caries prevention education	217	10	10	36	55	106
		(4.6%)	(4.6%)	(16.7%)	(25.3%)	(48.8%)
Education on oral hygiene measures	217	6	12	29	64	106
		(2.8%)	(5.5%)	(13.5%)	(29.4)	(48.8%)
Oral hygiene training	217	70	65	41	25	16
		(32.3%)	(30%)	(18.8%)	(11.5%)	(7.4%)
Nutritional counselling	217	13	11	34	69	90
		(6%)	(5.1%)	(15.6%)	(31.8%)	(41.5%)

**Table 6 children-09-01135-t006:** Fluoride recommendations of midwives in absolute values and in percentages (%).

	Total	Never	Rare	Occasionally	Often	Always
Fluoridated toothpaste from the first tooth	216	33	17	27	49	90
		(15.3%)	(7.9%)	(12.4%)	(22.7%)	(41.7%)
Fluoridated toothpaste, only if spitting out is possible	213	101	26	26	28	32
		(47.4%)	(12.2%)	(12.2%)	(13.2%)	(15%)
Fluoride-free toothpaste	214	103	34	37	19	21
		(48.1%)	(15.9%)	(17.3%)	(8.9%)	(9.8%)
Recommendation of fluoride tablets	217	119	40	22	16	20
		(54.8%)	(18.4%)	(10.2%)	(7.4%)	(9.2%)

## Data Availability

Not applicable.

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
