# Peer review of "Attitude of Midwives towards Fluoride Recommendations and Oral Prevention in Infants and Young Children"

_children, 2022, doi:10.3390/children9081135_

Round 1

Reviewer 1 Report

Table 1 in the Introduction might not be necessary since it could be added another sentence in lm 66 about oral hygiene instruction and fluoride application

Results ln 141-2 as well as Fig 1 ln 1, it was not clear whether all the 13220 registered midwives were recuited for this study, if not it should mention how the 1702 questionnaires were sampling. It would be helpful if there would be a table showing all the 14 questions of the questionnaires.

ln 128-9 mentioned that using Likert scale 1-5 for the questionnaires but Table 3 & 4 showed only 4 scales whereas Table 5 & 6, which was not consistency with the Likert 1-5 scale.

Discussion ln 2226 & 231 mentioned This study which confused wjether it meant for this paper but actually the authors meant for the references' study, please correct. ln 240 mentioned 5 names of the refrence authors which was not necessary. ln 286, the response rate was low (1.61%) which might not correct depending on the sampling methods of the study, not necessary using all registered midwives, as discussed earlier. There should be more discussion mentioning about the timing of this study after the adapting and changing recommendation between dentists and pediatricians in Germany which effected some result of this study such as the fluoride tablet used by midwives from this study. 

Reference 9 was an abstract

There were some English term that should be checked such as 'multipliers' (ln 303), '(dental) healthy nutrition (ln 56), 'dental application of fluoride' ln 70, etc.

Author Response

We thank the editors and reviewers for their thoughtful and helpful comments and

Suggestions and for the time and effort spent on them. Enclosed you will find our detailed point-by-point response.

We have revised the manuscript accordingly and highlighted all changes. We hope that our manuscript has been significantly improved and that the revised version is suitable for publication. As suggested, we used the help of the mdpi Author Services to improve the English language within our manuscript.

Reviewer 2 Report

I find the topic very interesting and of great importance for oral prevention. I congratulate the authors for the work on the originality and importance of their study, however, they present only a descriptive analysis with a questionable sample size.

Introduction: Although the authors make a good introduction to the state of the art in relation to early childhood caries and its impact on oral and general health, the title of the article focuses (or makes the reader think) on midwives and this is not reflected in the introduction. The authors need to rethink the way they should present it and explain more strongly the importance of midwives in infant oral health care.

Line 54-56 “Unfortunately, not enough pregnant women attend dental check-ups and are not aware of the importance of oral health [9]. Furthermore, parents of young children do not have sufficient knowledge about (dental) healthy nutrition and oral higiene”. The authors should add more references to support these claims.

 Line 55 ¿hygiene?

Line 57-66 references are missing.

Line 75-80 references are missing.

Materials and Methods:

2.1 Study Population: The authors must fully explain the characteristics of the population from which they took their sample, they only explain that they are from an association, but this is insufficient information.

Isn't there some sample size calculation? Or was it a census?

2.2. Questionnaire: Being a new questionnaire, the authors must present the questions to know them. 

Results:

Table 2. I do not see the need to present the gender row, if they are all women, this should be described in the description of the study population section.

Author Response

We would like to thank the editors and reviewers for their thoughtful and helpful comments and suggestions and for the time and effort spent on them.

Please find attached our detailed point-by-point response to the comments. We have revised the manuscript accordingly, highlighting all changes, and hope that the revised version is suitable for publication. As suggested, we used the help of the mdpi Author Services to improve the English language within our manuscript.

Round 2

Reviewer 2 Report

The authors have adequately addressed several of the concerns, however, they have not addressed the two main ones: they present only a descriptive analysis with a questionable sample size.

Regarding the explanation of the authors about the study population: "Unfortunately, we did not get information about the demographic data of the survey caregivers. The chosen study design is described under Material and Methods. It differs from a "classic questionnaire study". The study was conducted digitally, which means that no paper-based questionnaires had to be distributed by mail or at a conference or meeting. Instead, an online questionnaire served as the survey instrument. On the one hand, digital data collection reduces the risk of transmission errors that can occur when paper questionnaires are transferred to a database. On the other hand, it is an economical and time-saving method, since there are no printing or postage costs and there is no need to wait for returns. Another advantage is that the online participant does not feel observed and can complete the questionnaire with less social pressure and under anonymous conditions (data protection). However, due to German data protection and the anonymous study design (approved by the local ethic commission), it was not possible to contact midwives who did not respond to the call or participants who dropped out after the introductory page". All of this should be widely discussed and placed in the discussion. None of these problems and major sources of bias are addressed in the limitations section.

Author Response

We would like to sincerely thank the reviewers and editors for their important comments and suggestions for improving the manuscript and for the time they took.

Please find attached our detailed point-by-point response to the comments. We have revised the manuscript accordingly, highlighting all changes, and hope that the revised version is suitable for publication.
